

# Stabilizer disentangling of conformal field theories

**Martina Frau[1⋆], Poetri Sonya Tarabunga[2,3,4], Mario Collura[1,5],**
**Emanuele Tirrito[4,6] and Marcello Dalmonte[4]**

**1** International School for Advanced Studies (SISSA), Via Bonomea 265, I-34136 Trieste, Italy
**2** Technical University of Munich, TUM School of Natural Sciences,
Physics Department, 85748 Garching, Germany
**3** Munich Center for Quantum Science and Technology (MCQST),
Schellingstr. 4, 80799 München, Germany
**4** The Abdus Salam International Centre for Theoretical Physics (ICTP),
Strada Costiera 11, 34151 Trieste, Italy
**5** INFN, Sezione di Trieste, Via Valerio 2, 34127 Trieste, Italy
**6** Dipartimento di Fisica "E. Pancini", Università di Napoli "Federico II",
Monte S. Angelo, 80126 Napoli, Italy

⋆ mfrau@sissa.it

## Abstract

**Understanding how entanglement can be reduced through simple operations is crucial for both classical and quantum algorithms. We investigate the entanglement properties of lattice models hosting conformal field theories cooled via local Clifford operations, a procedure we refer to as stabilizer disentangling. We uncover two distinct regimes: a constant gain regime, where disentangling is volume-independent, and a log-gain regime, where disentanglement increases with volume, characterized by a reduced effective central charge. In both cases, disentangling efficiency correlates with the target state magic, with larger magic leading to more effective cooling. The dichotomy between the two cases stems from mutual stabilizer Rényi entropy, which influences the entanglement cooling process. We provide an analytical understanding of such effect in the context of cluster Ising models, that feature disentangling global Clifford operations. Our findings indicate that matrix product states possess subclasses based on the relationship between entanglement and magic, and clarifying the potential of new classes of variational states embedding Clifford dynamics within matrix product states.**

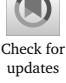

# 1   Introduction

Building on the revolutionary concept of formulating the renormalization group based on eigenvalues of reduced density matrices [1], tensor network (TN) states [2–8] have found widespread applications in the context of many-body theory. One key aspect of such class of variational state is that their limiting factor, entanglement, is very well understood, so that it is possible to exploit this fact for controlled computations. Over the years, there has been a growing interest in expanding the representation power of tensor network states, by dressing their structure with additional operations that could cope with such a limiting resource. One of the earliest examples along this line is the multi-entanglement renormalization Ansatz [9], where layers of additional generic entangling gates provide the entangling power. More recent examples are augmented tree tensor networks (TTN) [10,11] and augmented matrix product states (MPS) [12]. The basic idea behind the identification of larger basins of variational states is that one feeds a controlled amount of entanglement by dressing input states via (entangling) two-body operations. While very powerful in principle, such generic dressing of tensor networks often incurs in considerable computational overheads, and does not lend itself to simple physical interpretations in terms of which classes of states can be accessed.

A sharply different approach has been put forward recently, merging entanglement with another fundamental proxy of quantum complexity originating from quantum error correction – non-stabilizerness, also known as magic [13–15]. The basic idea is to dress tensor network states not with generic operations, but rather, with a restricted class of gates harvested from

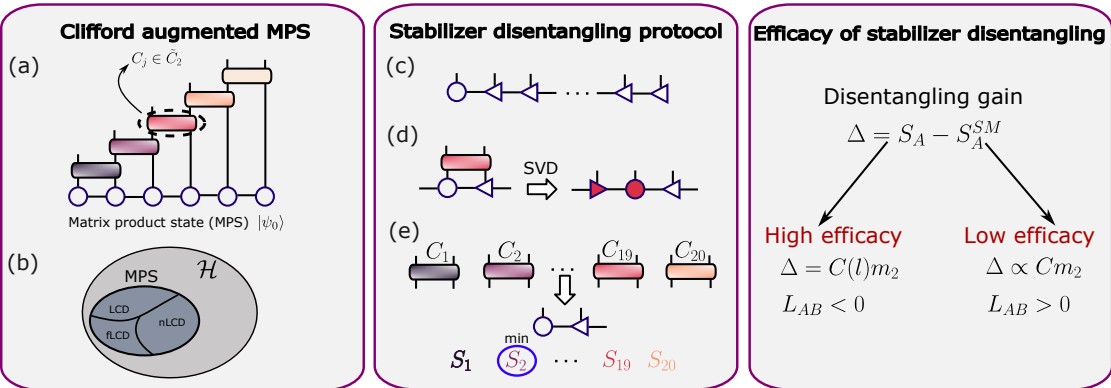

Figure 1: **a-b**: Clifford augmented MPS. **(a)**: schematics of a single sweep of local stabilizer disentangling protocol applied onto a matrix product state (MPS); **(b)**: Hilbert space structure of many-body steates for a spin system. The manifold comprising MPS can be further split with respect to how much they can be disentangled by local Clifford: local-Clifford disentanglable (LCD; disentangling improves with size), non local-Clifford disentanglable (nLCD; disentangling does not improve with size), and fully disentanglable (fLCD). In the text, we elaborate examples of all three cases: tricritical Ising point, XXZ critical, and cluster Ising model. **c-e**: Illustration of the Stabilizer disentangling protocol (see Sec.3.1). **(c)**: Obtain the groundstate with DMRG and bring it to a right-normalized form; **(d)** Apply the gates in $\tilde{\mathcal{C}}_2$ (see 3.1) to the first bond, selecting the gate that minimizes the entanglement entropy on that bond; **(e)** Apply the chosen gate to the MPS, perform a Singular Value Decomposition, update the first two tensors in the MPS, and repeat step (d) on the next bond.

the Clifford group: in one-dimension, the corresponding variational class goes under the name of Clifford augmented matrix product states (CAMPS) [16] or Stabiliser TN [17] (as well as the corresponding dressed operators [18]), that have been proposed inspired by the observation of strong relationship between entanglement and magic in both random [19] and ground matrix product states [20]. As far as the Clifford-dressing operation keeps a local structure, the new Ansatz allows to extend computations straightforwardly and, at the same time, to represent states with arbitrary entanglement with modest computational resources, as demonstrated within several algorithmic instances [16, 21–23]. The limiting factor of such approach is how can one represent a state at fixed magic, while lowering entanglement with strictly local operations. Understanding the descriptive power of such Clifford-augmented TN raises a deep question in terms of the entanglement/magic interplay in many-body states, that is, how can we actually "cool" their entanglement properties via application of stabilizer gates that are local in nature.

Here, we study the question whether one-dimensional conformal field theories [24] can be entanglement-cooled via local stabilizer operations. Such theories represent an ideal testing ground to understand the nature of stabilizer disentangling, as they combine a fundamental theoretical relevance, with a mild, logarithmic violation of the area law [25,26]. Starting from the matrix-product state representation of ground states of spin chains, we apply a Clifford circuit whose structure is made of local two-body gates. In order to comply with the structure of MPS, the basic protocol we follow is a sweeping procedure, schematically illustrated in Fig. 1a, where each two-body gate is harvested from the Clifford group on nearest-neighbors.

We find that the Hilbert space of lattice conformal field theories features three distinct classes, depending on how effectively local stabilizer operators disentanglement works (see Fig. 1b). The first class is made of states whose entanglement can only be lowered by a con-

stant: we deem those states non local-Clifford disentanglable (nLCD). The second class comprises states whose entanglement can be systematically cooled as size increases: the effective description of such systems is via a effective central charge, that is smaller with respect to the target state one. We define these staes are local-Clifford disentanglable (LCD). Finally, some states can be perfectly disentangled, at least for specific partitions, via local Clifford - we deem those fully LCD (fLCD).

A key result of our work is the close connection between magic and the efficiency of stabilizer disentangling. We find that the total magic content strongly correlates with, and in some cases is directly proportional to, the ability of local stabilizer operations to reduce entanglement. We argue how such - in principle, very unexpected - finding can be justified based on the very special correlation structure of ground states, that is at odds with that of random Haar states. Secondly, we show how the efficacy of local stabilizer cooling with respect to size is dictated by the mutual stabilizer Rényi entropy (mSRE) [27–29]. This quantity is akin to mutual information in the context of Rényi entropies, and describes how spread in stabilizer space a given partition is. Our results show that when mSRE is negative, stabilizer disentangling improves with system size, as seen in LCD and fLCD states. Conversely, when mSRE is positive, disentangling ceases to improve with size, which is characteristic of nLCD states (see Fig. 1f). We elucidate this fact via extensive numerical computations, and analytical understanding via exactly soluble cluster-Ising models [30]. Lastly, we show that the efficiency of disentangling is also related with a more refined measure of magic constrained by bond dimension. To probe this connection, we introduce an additional observable, $m_2^{\chi=2}$, which quantifies the magic of a state when restricted to a low-entanglement, limited bond dimension description. This quantity provides insight into the portion of magic that can be removed solely through local operations. Numerically, we establish a direct relationship between $m_2^{\chi=2}$ and the classification of states in terms of disentangling power. We find that when $m_2^{\chi=2}$ is much smaller than the magic of the full representation of the state, the state belongs to the non-local-Clifford disentanglable (nLCD) class. On the other hand, when the two magic measures are comparable, the state is local-Clifford disentanglable (LCD).

Our results thus show how, within the Hilbert space corner described by matrix product states, there is fundamental a difference between states that can be disentangled by Clifford operations, and states that cannot (see Fig. 1). This emphasises the potential for CAMPS to describe efficiently a broader class of phenomena with respect to MPS, with emphasis on states with vanishing non-local magic content.

The rest of the paper is structured as follows. In Sec. 2, we review the concept of magic, introduce the main observables we target, and review how those are computed in tensor network states. In Sec. 3, we describe the local entanglement cooling protocol, and show how CAMPS can be dramatically simplified. Sec. 4, we review the models we investigate numerically, while in Sec. 5, we discuss the corresponding numerical results. The origin of efficient stabilizer disentangling is then elaborated upon in Sec. 6, investigating cluster Ising models. Finally, we draw our conclusions and present a detail outlook of our findings in the broader context of augmented tensor network states.

# 2 Basics of magic

## 2.1 Definition

Magic quantifies the extent to which a quantum state deviates from the class of stabilizer states, a subset of quantum states that can be efficiently simulated on classical computers. In this section, we briefly review the key definitions necessary for their characterization.

Consider a system of $N$ qbits, with Hilbert space $\mathcal{H} = \otimes_{j=1}^{N} \mathcal{H}_i$. The Pauli group $\mathcal{P}_N$ is defined as

$$\mathcal{P}_N = \left\{ e^{i\theta \frac{\pi}{2}} \sigma_{j_1} \otimes \cdots \otimes \sigma_{j_N} | \theta, j_k = 0, 1, 2, 3 \right\}. \tag{1}$$

It includes all operator strings composed of tensor products of $N$ Pauli operators, each multiplied by a phase factor of $\pm 1$ or $\pm i$.

The stabilizer group $\mathcal{S}$ of a $N$-qbits pure quantum state $|\psi\rangle$ is an Abelian subgroup of $\mathcal{P}_N$ and contains all the strings that stabilize the state, namely

$$S|\psi\rangle = |\psi\rangle, \quad \forall S \in \mathcal{S}. \tag{2}$$

A state is a stabilizer states if $|\mathcal{S}| = 2^N$. Such a state can be efficiently simulated via stabilizer formalism [31]. Alternatively, a stabilizer state can be described in terms of the resources required for its preparation. The Clifford group of $N$ qbits $\mathcal{C}_N$ consists of unitary operators that serve as the normalizer of $\mathcal{P}_N$, mapping a Pauli string in another Pauli string via conjugaiton, specifically

$$\mathcal{C}_N = \left\{ U \text{ such that } UPU^\dagger \in \mathcal{P}_N \text{ for all } P \in \mathcal{P}_N \right\}. \tag{3}$$

It is generated by the Hadamard gate, the $\frac{\pi}{4}$ and the CNOT gate. It is well-known that Clifford group alone is insufficient for universal quantum computation [14], and must be supplemented by the T gate. Stabilizer state are those states that can be prepared by means of only Clifford operations, starting from a product state $|0\rangle^{\otimes N}$.

Magic represents the deviation from the set of stabilizer states, reflecting the amount of non-Clifford resources required for state preparation. For this reason, the properties required for a good measure of magic, that we denote as $\mathcal{M}$, are the following: (i) Faithfullness: $\mathcal{M}(|\psi\rangle) = 0$ iff $|\psi\rangle$ is a stabilizer state (ii) Invariance under Clifford unitaries: $\mathcal{M}(\Gamma|\psi\rangle) \leq \mathcal{M}(|\psi\rangle)$, for $\Gamma \in \mathcal{C}_N$, and (iii) Additivity: $\mathcal{M}(|\psi\rangle \otimes |\phi\rangle) = \mathcal{M}(|\psi\rangle) + \mathcal{M}(|\phi\rangle)$ (in general, sub-additivity is sufficient). Several measures that meet these criteria have been proposed in quantum information theory, including the min-relative entropy of magic [32], the relative entropy of nonstabilizerness [13], the robustness of magic [32,33], the stabilizer rank, stabilizer extend [34], mana, thauma (for qudit) [35], stabilizer nullity [36], CSS entropy [37], and the basis-minimized stabilizerness asymmetry [38]. However, many of these measures involve complex minimization procedures, which complicate their computation in many-body systems. To face the problem of computability, more practical measures of nonstabilizerness have been recently introduced, including Bell magic [39] and stabilizer Rényi entropies (SREs) [27]. Unfortunately, their calculation still requires evaluating an exponential number of terms. Nevertheless, notable progress has been made in their experimental measurements [40–43] and large-scale numerical simulations [28,29,44–48].

## 2.2 Full-state magic

In this study, we measure magic using SREs [27]. For a pure quantum state of $N$ qubits $\rho$, SREs are defined as

$$M_n(\rho) = \frac{1}{1-n} \log \left\{ \sum_{P \in \tilde{\mathcal{P}}_N} \frac{\text{Tr}(\rho P)^{2n}}{2^N} \right\}, \tag{4}$$

where $P$ is a Pauli string and $\tilde{\mathcal{P}}_N$ is the projective Pauli group, namely the standard Pauli group modulo the global phases. This definition satisfies properties (i), (ii) and (iii) mentioned in the previous section. Eq. 4 coincides with the $n$-Rényi entropy associated to the classical probability distribution

$$\Xi_P = \frac{\text{Tr}(\rho P)^2}{2^N}. \tag{5}$$

Magic generally scales linearly with the system size $N$. Therefore, in the following, we will consider the SRE density defined as $m_n = M_n/N$. The SREs provide useful measures for evaluating the total magic content of a quantum state. However, for the purpose of this work — specifically, assessing the effectiveness of the Stabilizer Disentangling Algorithm — metrics beyond the total magic of the full state are needed.

## 2.3 Mutual stabilizer Rényi entropy

We can extend the definition of SRE to mixed states, namely when $\mathrm{Tr}(\rho^2) \neq 1$, by properly normalizing the above probability,

$$\tilde{\Xi}_P = \frac{\mathrm{Tr}(\rho P)^2}{2^N \mathrm{Tr}(\rho^2)}, \tag{6}$$

such that we still have $\sum_P \tilde{\Xi}_P = 1$. For example, for the case of Rényi index $n = 2$, the SRE of a mixed state reads

$$\tilde{M}_2 = -\log\left(\frac{\sum_{P \in P_N} |\mathrm{Tr}(\rho P)|^4}{\sum_{P \in P_N} |\mathrm{Tr}(\rho P)|^2}\right). \tag{7}$$

With this extension, we define the mutual SRE as

$$L_n(\rho_{AB}) = \tilde{M}_n(\rho_{AB}) - \tilde{M}_n(\rho_B) - \tilde{M}_n(\rho_A). \tag{8}$$

This quantity, reminiscent of mutual information, was first introduced in Ref. [29], and subsequently investigated in [20, 49–51]. It is worth noting that, similar to mutual Rényi entropies, $L_n(\rho_{AB})$ is not necessarily strictly positive in general.

### 2.3.1 Interpretation of mutual SRE

It is important to note that, very much like Rényi mutual information in the context of entanglement (which is not positive definite), the mutual SRE entropy cannot be interpreted as a generic information measure, as the reduced density matrices $\rho_A, \rho_B$ are in general mixed. It is however very indicative of correlations, as already demonstrated in the context of quantum phase transitions, and again, in a similar manner as Rényi mutual information for entanglement.

We would like now to provide two arguments to justify its physical relevance, that will also be instrumental in interpreting the results presented below.

**Spreading of information over Pauli strings.** The first perspective we would like to offer is based on the fact that SRE can be thought of as participation entropy in the Pauli basis [46, 52]. This interpretation remains correct even in cases where SRE is not a measure of magic, for instance for mixed states.

The mSRE can then be thought of as follows: if positive, this indicates that the sum of the two partitions is actually more widely spread in stabilizer subspace with respect to the original partitions (taking into account the normalization factor at the denominator in Eq. (6)). Oppositely, if negative, it implies that there is some common stabilizer states the sum of the two partitions projects onto, so as to induce a specific form of correlations between them. In the context of stabilizer disentangling, the presence of such correlation will be related to a more efficient disentangling procedure: this may be expected based on the fact that, once the circuit has sufficiently large support to capture both $A$ and $B$, there is an additional degree of freedom/non-separability that can be taken care of solely via stabilizer operations.

We can have a qualitative understanding of the above argument in a simplified scenario. Suppose that there is a single stabilizer operator commuting with the state $\rho$, and with support on both $A$ and $B$. Then, we expect a negative (albeit small due to normalization) mSRE. We will see that this is exactly what happens in cluster Ising models, where the presence of such 'hidden' encoded qubit makes stabilizer disentangling extremely efficient.

**Density matrix decomposition.** The second argument is based on the fact that any mixed state on two disconnected partitions $A$ and $B$ can be written as follow

$$\rho_{AB} = \frac{1}{2^{N_B}}\rho_A I_B + \frac{1}{2^{N_A}}I_A\rho_B - \frac{1}{2^N}I_A I_B + \varrho_{AB}\,, \tag{9}$$

where the term proportional to the identity accounts for the correct normalization of the full operator, and we have $|A| = N_A$, $|B| = N_B$, $N_A + N_B = N$. The previous equation implicitly defines the last term $\varrho_{AB}$ which incorporates all contributions coming from Pauli strings which have nontrivial support in both subsystems $A$ and $B$. As a consequence, $\text{Tr}(\varrho_{AB}) = 0$. In particular, when this term is absent, it is easy to see that the SREs of $AB$ reduces to a particular combination of the SREs of $A$ and $B$ separately. In fact, assuming $\varrho_{AB} = 0$ one gets for example (for $n = 2$)

$$\text{Tr}(P\rho_{AB})^4 = \delta_{P_B I_B}\text{Tr}(P_A\rho_A)^4 + \delta_{P_A I_A}\text{Tr}(P_B\rho_B)^4 - \delta_{PI}\,, \tag{10}$$

where $P = P_A P_B$ with $P_A(P_B)$ is acting in the subsystem $A(B)$ only. It follows that all Pauli strings with non-trivial support in both $A$ and $B$ have vanishing expectation values. As a qualitative consequence, the mutual SRE would be particularly sensible to the absence/presence in the full density matrix $\rho_{AB}$, of Pauli correlations acting non-trivially in $A$ and $B$. In fact, after subtracting from $L_2(\rho_{AB})$ the entanglement contribution (namely the Rényi-2 mutual information $I(\rho_{AB})$), the remaining "magic" part $W(\rho_{AB})$ (see Eq.(12)) strongly depends on $\varrho_{AB}$.

## 2.4 Magic that can be removed by local operations

Another useful observable that we will consider below attempts to quantify the magic that a given state has if restricted to a limited bond dimension description. This quantity is related to the portion of magic that can be removed by applying only local operations on the state. While the detailed reasoning behind introducing this quantity, that we denote as $m_2^{\chi=2}$, will be thoroughly discussed in Sec. 2.5, we anticipate an intuitive explanation of this measure here: $m_2^{\chi=2}(|\psi\rangle)$ represents the magic of the state closest (in $L_2$ norm) to $|\psi\rangle$ among those that exhibit low entanglement. If the magic is primarily derived from the correlations we will get $m_2^{\chi=2} \ll m_2$. Conversely, if $m_2^{\chi=2} \sim m_2$, this indicates that most of the state's magic content is not due to correlations and can therefore be eliminated through local operations. In this scenario, we would expect the Stabilizer Disentangling Protocol to be more effective.

## 2.5 Numerical computation

Our computations are all based on MPSs, the simplest class of Tensor Network states, a very established and powerful tool to describe groundstates of 1$D$ spin-chains [53]. A pure quantum state $|\psi\rangle$, in its MPS reperesentation, reads

$$|\psi\rangle = \sum_{s_1,s_2,\ldots,s_L} A^{s_1}A^{s_2}\cdots A^{s_L}|s_1,s_2,\ldots s_L\rangle\,, \tag{11}$$

where $A^{s_i}$ are $\chi \times \chi$, except for the boundaries $i = 1$, $i = L$, where the corresponding tensors are, respectively, a row and a column vector of length $\chi$. The parameter $\chi$ is the *bond dimension* of the MPS and it's directly related to the entanglement of $|\psi\rangle$.

By utilizing this representation of the state and incorporating sampling techniques within the space of the Pauli strings, we can efficiently derive the magic observables defined earlier. Indeed, while computations for generic states typically scale exponentially with $N$, this framework allows us to reduce the computational complexity to a linear scaling with respect to system size $L$ and number of samples $N_S$, and polynomial in the bond dimension $\chi$.

Specifically, for full state magic density $m_n$ we exploit perfect Pauli sampling, a method based on Tensor Networks perfect sampling algorithms [54,55], and introduced for the computation of magic in Ref. [44,45]. We employ the same technique also for the computation of $m_2^{\chi=2}$. In this case, we first variationally project the MPS $|\psi\rangle$ onto the manifold of $\chi = 2$ MPSs, and then repeat the computation. We choose $\chi = 2$ because it is the smallest bond dimension that exhibits a non-pathological and non-trivial behavior. In fact, the choice $\chi = 1$ corresponds to product states, and the projection onto this manifold is too abrupt and leads to an unpredictable behavior of magic. Furthermore, some of the phases present in the model we are considering have a fixed point that can be exactly captured with a bond dimension of 2, making it crucial to retain those correlations in our analysis.

One of the limitations of Perfect Pauli sampling is that it does not allow for computation mutual SRE between two disconnected partitions $A$, $B$ as in Eq. 8. To obtain $L_n(\rho_{AB})$ we employ the Pauli-Markov sampling method introduced in Ref. [29]. In particular, we focus on the case $n = 2$, from now on denoted just $L(\rho_{AB})$, and rewrite $L(\rho_{AB})$ as

$$L(\rho_{AB}) = I(\rho_{AB}) - W(\rho_{AB}), \tag{12}$$

where $I(\rho_{AB}) = S_2(\rho_A) + S_2(\rho_B) - S_2(\rho_{AB})$ is the Rényi-2 mutual information, and

$$W(\rho_{AB}) = -\log\left(\frac{\sum_{P_A \in \mathcal{P}_A} |\mathrm{Tr}(\rho_A P_A)|^4 \sum_{P \in \mathcal{P}_B} |\mathrm{Tr}(\rho_B P_B)|^4}{\sum_{P_{AB} \in \mathcal{P}_{AB}} |\mathrm{Tr}(\rho_{AB} P_{AB})|^4}\right).$$

Each of the two terms in Eq. 12 can be sampled via a Markov-chain approach, as detailed in Ref. [29].

# 3 Stabilizer disentangling via Clifford

## 3.1 Stabilizer disentangling protocol

We perform a disentangling procedure on a given ground state, previously obtained via standard DMRG, using a Clifford circuit consisting of two-qubits Clifford operations. The basic steps of the algorithm are depicted in Fig. 1c-e, and can be listed as follows:

(i) given the target Hamiltonian $H$, we run a DMRG simulation to obtain the ground state $\Psi$ in MPS form;

(ii) we apply all possible two-qubit gates on the bond $(1,2)$, obtaining the (unnormalized) states $|\Psi[C_1]\rangle = C_{1,2}|\psi\rangle$;

(iii) based on their bipartite entanglement entropy at the cut $(1,2)$, we select the lowest entangled state within the set of all $|\Psi[C_1]\rangle$; we denote this state as $|\Psi[C_1^m]\rangle$, and the corresponding minimizing gate as $C_{1,2}^m$;

(iv) we repeat operations (ii-iii) sequentially for all bonds, until one obtains the CAMPS:

$$|\Psi[C_{1;L-1}]\rangle = \prod_{j=1}^{L-1} C_{j,j+1}|\psi\rangle, \tag{13}$$

whose entanglement is smaller than $\Psi$;

(v) repeat (ii-iv) until convergence with the number of sweeps.

The resulting CAMPS will have the structure as shown in Fig. 1a. By construction, the MPS part of the CAMPS will have a systematically lower entanglement entropy compared to the MPS representation of the initial state. Note that, because of the circuit structure, the circuit becomes global after just one sweep with $O(N)$ gates. Namely, a local operator can be transformed to an operator with support on an extensive number of qubits. This is different from, e.g., brickwork circuit, where $O(N)$ depth and $O(N^2)$ gates are required.

**Optimized search.** In the original CAMPS proposal [16], the full set of two-qbits Clifford gates (a part from phase factors) comprising 700+ gates was used. Here, we propose a largely optimized search. Let $\mathcal{C}_2$ denote the original set of two-qubit gates and $\mathcal{C}_1$ denote the set of single-qubit Clifford gates. The key observation is that if two unitaries $U, V \in \mathcal{C}_2$ can be expressed as $U = (S_1 \otimes S_2)V$, where $S_1, S_2 \in \mathcal{C}_1$, then $U$ and $V$ will have identical effects on the singular values. Therefore, the search for disentangling can be restricted to a subset of $\mathcal{C}_2$, that we denote $\tilde{\mathcal{C}}_2$, consisting of $|\mathcal{C}_2|/|\mathcal{C}_1 \otimes \mathcal{C}_1| = 20$ gates.

## 3.2 Entropy gain $\Delta$

To evaluate the representational capacity of Clifford-augmented tensor networks, it is essential to define an observable that quantifies the efficiency of the Stabilizer disentangling protocol. We refer to this observable as entropy gain. For a subregion $A$ of size $\ell$, it is defined as

$$\Delta = S_A - S_A^{SM}. \tag{14}$$

Here, $S_A$ denotes the entanglement entropy of the original state $|\Psi\rangle$, while $S_A^{(SM)}$ represents the entanglement entropy of the resulting state after applying the stabilizer disentangling protocol, specifically

$$S_A^{(SM)} = -\mathrm{Tr}\rho_A^{(SM)} \ln \rho_A^{(SM)},$$

where

$$\rho_A^{(SM)} = \mathrm{Tr}_{\bar{A}} |\Psi[C_{\ell;L-\ell}]\rangle \langle \Psi[C_{\ell;L-\ell}]|$$

is the reduced density matrix of the final state, with the trace taken over the complement of subregion $A$. We will refer to $S_A^{(SM)}$ as Stabilizer Minimized Entanglement Entropy (SMEE).

# 4 Models

In this section we provide a summary of the properties of one-dimensional models employed to explore stabilizer disentangling. For each of the following models, we obtain the MPS approximation of the ground state at a specified bond dimension, $\chi$, by performing Density Matrix Renormalization Group (DMRG) simulations using the iTensor package [56, 57]. We made sure that the results have converged in the bond dimension, so that the MPS is a faithful representation of the ground state. Subsequently, we apply the techniques described in the previous section to calculate the quantities of interest from the MPS.

Our focus in the following will be on critical spin chains governed by conformal field theory, where the entanglement entropy is known to exhibit the scaling behavior [26, 58]

$$S_A = \frac{c}{6} \log\left[\frac{2L}{\pi} \sin\left(\ell\frac{\pi}{L}\right)\right] + \gamma, \tag{15}$$

in open chains. Here, $\ell$ denotes the subsystem size, $L$ is the total size, $c$ is the central charge of the conformal field theory, and $\gamma$ is a non-universal constant. The scaling behavior above

implies that a polynomial growth of bond dimension $\chi$ with $L$ is necessary for an MPS to faithfully represent the state. In the following we will examine how the stabilizer disentangling procedure affects the CFT entanglement scaling.

## 4.1 XXZ model

We consider a spin-$\frac{1}{2}$ XXZ chain with open boundary conditions, described by the Hamiltonian:

$$H = -\sum_{i=1}^{L-1}\left(S_i^x S_{i+1}^x + S_i^y S_{i+1}^y + J_z S_i^z S_{i+1}^z\right), \tag{16}$$

where $S_i^\alpha = \frac{1}{2}\sigma_i^\alpha$ and $\sigma_i^\alpha$ are the Pauli matrices (with $\alpha = x, y, z$). $J_z$ is the parameter that represents the strength of the uniaxial anisotropy along the $z$-direction.

The model exhibits three distinct phases [59]: A gapped ferromagnet for $J_z > 1$; a Tomonaga-Luttinger liquid phase, characterized by a CFT with a central charge $c = 1$ in the interval $|J_z| < 1$; A gapped antiferromagnet for $J_z < -1$. For $|J_z| = 1$ the model becomes equivalent to the Heisenberg model (After rotation along $z$ when $J_z = -1$). In what follows, we will consider groundstates of this model inside the critical phase.

## 4.2 Tricritical Ising model

We also examine a quantum spin chain with a tricritical Ising point, as first introduced in Ref. [60]. The Hamiltonian for this model is $\mathbb{Z}_2$ symmetric and is given by:

$$H = H_I + \lambda H_T, \tag{17}$$

where $H_I$ represents the transverse field Ising model hamiltonian at the critical point $h = 1$, specifically

$$H_I = -\sum_{i=1}^{L-1} S_i^z S_{i+1}^z - \sum_{i=1}^{L} S_i^x, \tag{18}$$

and $H_T$ describes a three-spins interaction term:

$$H_T = \sum_{i=1}^{L-2}\left(S_i^x S_{i+1}^z S_{i+2}^z + S_i^z S_{i+1}^z S_{i+2}^x\right). \tag{19}$$

The model remains in the Ising universality class for $0 \leq \lambda < \lambda_{\text{TCI}}$, with a tricritical Ising (TCI) point at $\lambda_{\text{TCI}} = 0.428$, marking the transition between the critical Ising phase and a gapped phase. In the gapped phase, a $\mathbb{Z}_2$ symmetry-preserving ground state coexists with two symmetry-broken ground states. In the following, we focus on the TCI point in the phase diagram, which is described by a conformal field theory with central charge $c = 7/10$.

## 4.3 Cluster Ising models

Finally, we consider a class of models known as Cluster Ising models [30, 61–63], focusing on three distinct scenarios with corresponding Hamiltonians $H_1$, $H_2$, and $H_3$. The first Hamiltonian is defined as follows

$$H_1 = -\sum_{i=1}^{L-2}(S_i^x S_{i+1}^z S_{i+2}^x) + h\sum_{i=1}^{L} S_i^z. \tag{20}$$

Using a Jordan-Wigner transformation, Ref. [63] demonstrates that the model exhibits a phase transition at $h = 1$. For $h = 0$, the system's four-fold degenerate ground state (in case of open

boundary conditions) is the cluster state, an example of symmetry-protected topologically ordered state, protected by a $\mathbb{Z}_2 \times \mathbb{Z}_2$ symmetry, which has been extensively studied as a resource for quantum computation [64, 65].

Expanding on this model, we introduce an additional nearest-neighbor interaction, weighted by the parameter $D$, as shown in the following Hamiltonian

$$H_2 = H_1 + D \sum_{i=1}^{L-1} S_i^z S_{i+1}^z. \tag{21}$$

Details of our computation of the critical points for this model can be found in Appendix B. Our estimate for the critical point at $D = 0.1$ yield $h_c \simeq 0.9$.

In both the previous cases, the critical points of the phase diagram are described by a Conformal Field theory with central charge $c = 1$.

The third and last scenario we examine, presented in detail in Ref. [30], is the following:

$$H_3 = -\sum_{i=1}^{L-2} (S_i^x S_{i+1}^z S_{i+2}^x) + V \sum_{i=1}^{L-1} S_i^y S_{i+1}^y. \tag{22}$$

The phase diagram closely resembles that of the previous Cluster Ising model considered in this section. In fact, for $V = 0$, the two models coincide. The model host two phases: a cluster phase for $V < 1$, where multipartite entanglement is maximum, and, for $V > 1$, an antiferromagnetic phase, marked by nonvanishing staggered magnetization along the $y$-axis and a pronounced reduction in multipartite entanglement. At $V = 1$, the critical behavior of the model is described by a conformal field theory with central charge $c = 3/2$.

# 5 Results

To explore the relationship between properties of different states and the efficacy of Stabilizer Disentangling, we apply the stabilizer disentangling protocol in Sec.3.1 to various ground states. We then compute the magic observables defined in Sec.2 to examine correlations between the magic content of each state and its disentanglability.

In studying the representational power of TNs for many-body states, a key factor is the scaling of entanglement entropy with system size. It is well known that a MPS can efficiently approximate the ground state of a finite spin chain with $L$ sites when the entanglement entropy $S_A$ of a partition $A$ (comprising half the chain) is bounded by $S_A \leq c \log(L/2)$ [66]. To understand the impact of the Stabilizer Disentangling protocol on this bound, we analyze the size scaling of both entanglement entropy and SMEE.

We present the results organized by model: in Sec. 5.1 we analyze the XXZ model, and in Sec. 5.2 the Tricritical Ising model. In these cases, we observed that convergence showed no dependence on the number of sweeps: a single sweep was sufficient to achieve the minimum SMEE.

## 5.1 XXZ model

Ref. [67] demonstrates that Clifford circuits augmented with $T$-gates can be fully disentangled through Stabilizer Cooling, provided that the number of $T$-gates remains below the system size, $L$. The authors also delucidates that in the context of Hamiltonian dynamics, the behavior differs: even at low magic density, states exhibit resistance to stabilizer disentanglement, meaning they cannot be fully disentangled.

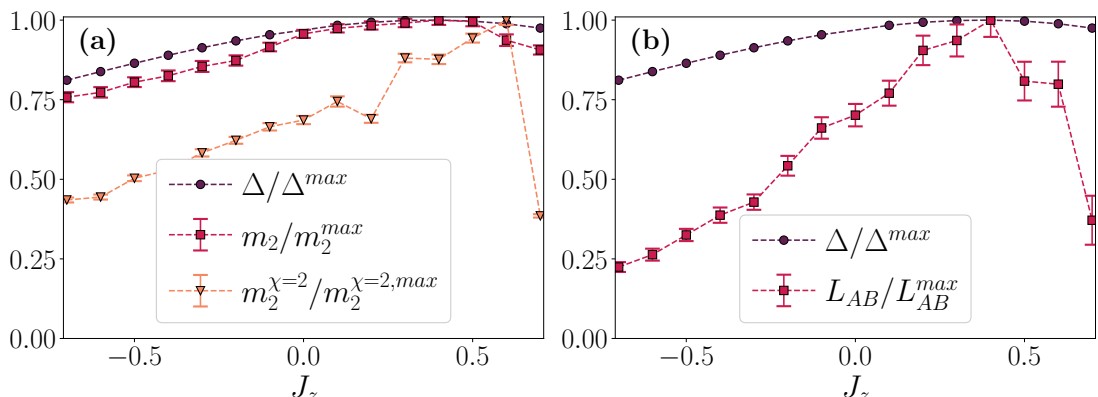

Figure 2: Entropy gain $\Delta$ compared with magic observables in the ground states of the XXZ chain across various $J_z$ values within the critical phase, for a system size $L = 32$. The data are presented on a relative scale, with each panel scaled as follows: **(a)**: $\Delta_{\max} = 0.67$, $m_2^{\max} = 0.27$, $m_2^{\chi=2,\max} = 0.30$; **(b)**: $\Delta_{\max} = 0.67$, $|L_{AB}^{\max}| = 0.27$. **(a)** Comparison with the full state SRE density $m_2$ and the local magic $m_2^{\chi=2}$. The computation of $m_2$ and $m_2^{\chi=2}$ is performed via perfect Pauli sampling with $N_S = 10^3$ samples. **(b)** Comparison with mSRE $L_{AB}$ where $A$, $B$ are two partitions located at the boundary of the chain, each of length $\ell_A = \ell_B = L/4$. The computation of $L_{AB}$ is performed via Pauli-Markov sampling with $N_S = 10^4$ samples.

Our findings for the critical groundstates of the XXZ model align with those results: these states cannot be disentangled completely, as shown in Fig. 3. However, we also found a new characteristic specific to Stabilizer disentangling of groundstates. Fig. 2 presents the entropy gain, $\Delta$, along with various magic observables within the critical phase of the XXZ model, with a fixed system size of $L = 32$. Interestingly, in Fig. 2a, we observe that

$$\Delta \sim C \, m_2 \,, \tag{23}$$

where $C$ is a proportionality constant dependent on system size, but not on the model's parameters.

Fig. 3 presents the results for the size scalings. In Fig.s 3a, 3b we observe the expected logarithmic scaling of entanglement entropy within the critical phase [26]. Specifically, Fig. 3a shows the scaling of $S_A$ for a fixed system size $L$ of $L = 512$ as the size $\ell$ of the partition $A$ varies. The $x$-axis is rescaled as $\log(\frac{2L}{\pi} \sin(\frac{\pi l}{L}))$. The dashed line in the figure corresponds to the linear fit: $S_A = a_1 \, \log(\frac{2L}{\pi} \sin(\frac{\pi l}{L})) + b_1$ (see Eq. (15)). The value of the central charge $c_1$ estimated using the relation $a_1 = \frac{c_1}{6}$ is found to be $c_1 = 1.016 \pm 0.001$. Figure 3b, on the other hand, displays the entanglement entropy $S_A$ calculated fixing the size of partition $A$ at $l = \frac{L}{2}$ and varying the size of the system $L$. In this case, the $x$-axis is rescaled as $\log(\frac{L}{\pi})$, and the central charge estimated by the fit $S_A = a_2 \, \log(\frac{L}{\pi}) + a_2$ (represented by the dashed line) is $c_2 = 1.015 \pm 0.001$. Both $c_1$ and $c_2$ are compatible with the central charge of the model that is $c = 1$.

In Fig. 3, we also present the results for the SMEE. As shown in Fig. 3a and Fig. 3b, the SMEE also follows the expected logarithmic trend, but appears lowered by a constant factor. Indeed, in Fig. 3c and 3d, where we illustrate the scaling of entropy gain $\Delta$ with system size, we observe that it scales sublogarithmically. Additionally, in both cases, the range of the $y$-axis is sufficiently narrow to be considered nearly constant, suggesting that the factor $C$ in Eq. 23 shows no size dependence in this context. We refer to states in which Stabilizer disentangling

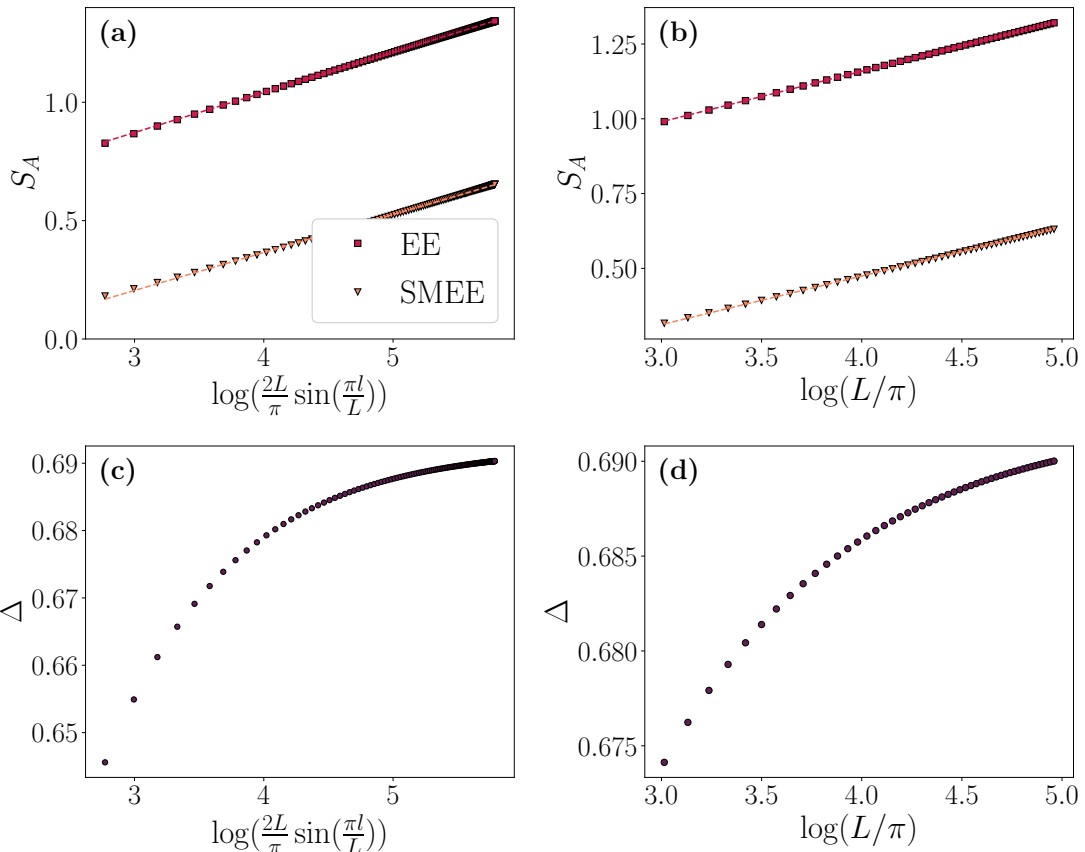

Figure 3: Stabilizer disentangling within the crical phase in XXZ model. For **(a)**, **(c)** data are shown for a fixes length of the system $L = 512$, varying the size of the partitions $A$, $B$ ($\ell_A = l$, $\ell_B = L - l$) from $l = 8$ to $l = 256$. For **(b)**, **(d)** data are presented for varying total system lengths $L$ with fixed, equally sized partitions $A$ and $B$ (where $\ell_A = \ell_B = L/2$). **(a)**, **(b)**: Scaling of Entanglement Entropy (EE) and Stabilizer-Minimized Entanglement Entropy (SMEE). Dashed lines represent the fitted curves, with both curves sharing the same slope. **(c)**, **(d)** Scaling of entropy gain $\Delta$. It exhibits sublogarithmic scaling; however, due to the very small scale on the y-axis, it can effectively be considered saturated.

does not improve with increasing system size as non-Local Clifford Disentanglable (nLCD) states, since the gain $\Delta$ becomes negligible in the large-$L$ limit. As previously discussed, this behavior is not governed by the full-state magic density $m_2$; instead, it is associated with the state's non-local magic content. To illustrate this fact, in Fig. 2, we consider $m_2^{\chi=2}$, which represents the amount of magic that cannot be eliminated by local operations. The latter is systematically smaller than $m_2$. This indicates that the magic primarily originates from the correlations within the state, making it more challenging to remove entanglement through stabilizer operations.

## 5.2 Tricritical Ising model

We now move to the Tricritical Ising Model. In Fig. 4, we present the comparison between the entropy gain $\Delta$ and the various magic observables throughout the phase diagram. Consistent with the results for the XXZ model, we find a correlation between the entropy gain, $\Delta$, and the

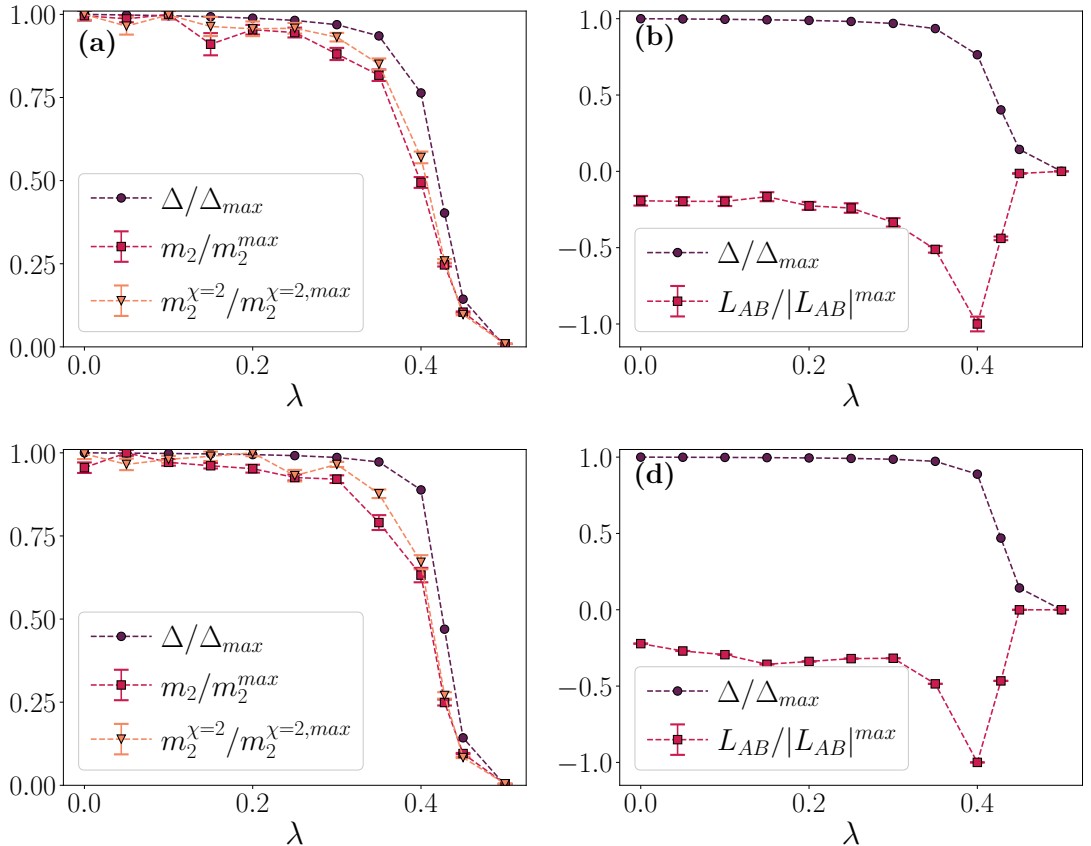

Figure 4: Entropy gain $\Delta$ compared with magic observables in the ground states of the Tricritical Ising model across various $J_z$ values, for two system sizes: $L = 64$ (**(a)** and **(b)**) and $L = 128$ (**(c)** and **(d)**). The data are presented on a relative scale, with each panel scaled as follows: **(a)**: $\Delta_{\max} = 0.34$, $m_2^{\max} = 0.29$, $m_2^{\chi=2,\max} = 0.29$; **(b)**: $\Delta_{\max} = 0.34$, $|L_{AB}^{\max}| = 0.003$; **(c)**: $\Delta_{\max} = 0.34$, $m_2^{\max} = 0.31$, $m_2^{\chi=2,\max} = 0.31$; **(d)**: $\Delta_{\max} = 0.34$, $|L_{AB}^{\max}| = 0.002$. **(a)**, **(c)**: Comparison with the full state SRE density $m_2$ and the local magic $m_2^{\chi=2}$. The computation of $m_2$ and $m_2^{\chi=2}$ is performed via perfect Pauli sampling with $N_S = 10^3$ samples. **(b)**, **(d)**: Comparison with mSRE $L_{AB}$ where $A$, $B$ are two partitions located at the boundary of the chain, each of length $\ell_A = \ell_B = L/4$. The computation of $L_{AB}$ is performed via Pauli-Markov sampling with $N_S = 10^4$ samples.

full state magic density, $m_2$, for both system sizes $L = 64$ (in Fig. 4a) and $L = 128$ (in Fig. 4b). Specifically, we find

$$\Delta \sim C \, m_2 \,. \tag{24}$$

What distinguishes this scenario from the previous one is the size scaling of EE and SMEE at the Tricritical Ising point $\lambda_{\mathrm{TCI}} = 0.428$, as illustrated in Fig. 5. In Fig.s 5a, 5b we observe that, when fixing the system size at $L = 512$ and varying the partition size $\ell$, or alternatively, fixing $\ell = \frac{L}{2}$ and varying $L$, both size scalings exhibit the expected logarithmic behavior. Specifically, the central charges obtained by the linear fits with the functional forms $S_A = a_1 \, \log(\frac{2L}{\pi} \sin(\frac{\pi l}{L})) + b_1$ and $S_A = a_2 \, \log(\frac{L}{\pi}) + b_2$ are $c_1 = 0.705 \pm 0.003$ and $c_2 = 0.736 \pm 0.005$, respectively, both of which are consistent with the central charge of the Tricritical Ising point, $c = 0.7$.

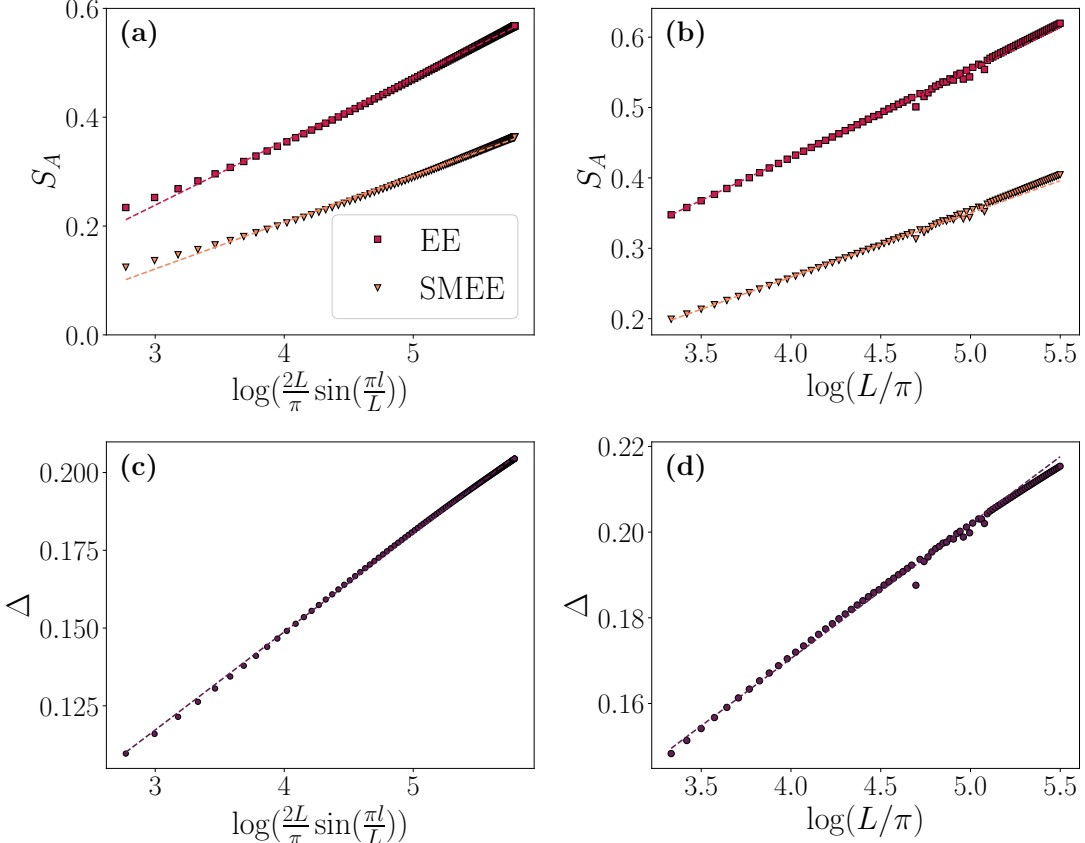

Figure 5: Entanglement entropy (EE) and Stabilizer-minimized Entanglement Entropy (SMEE) in Tricritical Ising point. For **(a)**, **(c)** data are shown for a fixed length of the system $L = 512$, varying the size of the partitions $A$, $B$ ($\ell_A = l$, $\ell_B = L-l$) from $l = 8$ to $l = 256$. For **(b)**, **(d)** data are presented for varying total system lengths $L$ with fixed, equally sized partitions $A$ and $B$ (where $\ell_A = \ell_B = L/2$). **(a)**, **(b)**: EE and CMEE scaling with size. While both follow the expected logarithmic trend, CMEE features a distinct prefactor. Dashed lines represent the linear fits. **(b)**, **(d)**: The gain $\Delta$ scales logarithmically with size at the TCI point. The dashed line represent the linear fit.

Most importantly, at odds with what happens for the XXZ model, we observe that SMEEs are not only shifted of a constant factor - they indeed show a logarithmic growth with a small prefactor. The estimated slope values are $a_1 = 0.086 \pm 0.001$ (from Fig. 5a) and $a_2 = 0.091 \pm 0.001$ (from Fig. 5b). This fact suggests that the Stabilizer disentanglement is able to map the model onto a different one, characterized by a smaller effective central charge $c^{SM}$. This behavior is further illustrated in Fig. 5c and Fig. 5d, where the logarithmic scaling of $\Delta$ with size is evident. The prefactors, determined by the fits (represented by the dashed lines) with the functional forms $\Delta = a_1^\Delta \log(\frac{2L}{\pi}\sin(\frac{\pi l}{L})) + b_1^\Delta$ for Fig. 5c and $\Delta = a_2^\Delta \log(\frac{L}{\pi}) + b_2^\Delta$ Fig. 5d, are $a_1^\Delta = a_2^\Delta = 0.03$, which correspond to a significant fraction of the central charge $c = 0.7$. We refer to conformal field theories for which $\Delta$ shows logarithmic scaling with system size as Local Clifford Disentanglable (LCD). What radically changes from nLCD states is the non-local magic. As shown in Fig. 4a, $m_2^{\chi=2} \sim m_2$, which implies that the magic of the state can be almost entirely removed by local operation. In this case, disentangling via Clifford operations is anticipated to be particularly effective. Our results confirm these expectations.

Furthermore, the distinction between nLCD and LCD conformal field theories can be understood through the mSRE of their corresponding states. In Fig. 2b, we see that in XXZ model $L_{AB} > 0$ within the critical phase. In contrast, Fig. 4b (for system size $L = 64$) and Fig. 4d (for system size $L = 128$) show that $L_{AB} < 0$ at $\lambda_{\text{TCI}} = 0.428$. As previously discussed in Sec. 2.3, we expected a more efficient Stabilizer disentangling procedure in the latter case, and our results confirm the prediction.

# 6 On the origin of efficient stabilizer disentangling: Cluster Ising models

While negative mSRE provides an intuitive way to understand why certain states are subjected to more efficient stabilizer disentangling than others, it is unable to provide clear information about the functional form, nor the origin of such a mechanism. Here, we show that both aspects can be addressed using Cluster Ising models as testing ground.

### 6.0.1 Non-local disentangling transformation as a Clifford operation

A key aspect of the cluster Ising models defined above is that, at their critical points, their ground states can be exactly mapped onto direct sums of lattice field theories. This fact, amply discussed in Ref. [30], can be understood in two ways. The first one is to note, as in Ref. [30], that the model free energy feature exact periodicity. The second one, more intuitive, is to re-write the model Hamiltonian in terms of complex fermions $c_j = \left(\prod_{i=1}^{j-1} \sigma_j^z\right)\sigma_j^-$, $c_j^\dagger = \left(\prod_{i=1}^{j-1} \sigma_j^z\right)\sigma_j^+$ (where $\sigma_j^\pm = \sigma_j^x \pm \sigma_j^y$) so that Eq. 22 becomes

$$H_3 = \sum_{i=1}^{L}(c_j^\dagger - c_j)(c_{j+2}^\dagger + c_{j+2}) + h \sum_{j=1}^{L}(c_j^\dagger + c_j)(c_{j+1}^\dagger - c_{j+1}).$$

By Fourier transforming the fermionic operators and applying a Bogoliubov transformation $b_k = u_k \gamma_k + i v_k \gamma_k^\dagger$ where $u_k = \frac{1}{\sqrt{2}}\sqrt{1 + \frac{\epsilon_k}{\Lambda_k}}$, $v_k = -\frac{1}{\sqrt{2}} \sin \delta_k \sqrt{1 - \frac{\epsilon_k}{\Lambda_k}}$ and

$$\epsilon_k = \cos\left(\frac{4\pi k}{L}\right) - h \cos\left(\frac{2\pi k}{L}\right),$$
$$\delta_k = \sin\left(\frac{4\pi k}{L}\right) + h \sin\left(\frac{2\pi k}{L}\right),$$

and, finally, $\Lambda_k = \sqrt{1 + h^2 - 2h\cos(\frac{6\pi k}{L})}$, the Hamiltonian becomes

$$H_3 = 2\sum_{k=1}^{L} \Lambda_k \left(\gamma_k^\dagger \gamma_k - \frac{1}{2}\right). \tag{25}$$

From this form, by choosing $M = \frac{L}{3}$ and splitting the summation into three separate terms, we express the Hamiltonian as

$$H = \sum_{s=1}^{3} H_{\text{Ising}}^s, \tag{26}$$

thanks to the fact that by inserting $M$ in $\Lambda_k$ the dispersion relation of the Ising model is retrieved. An analogous computation can be carried out for the Hamiltonian of Eq. 20, where the absence of nearest-neighbour interaction in the $y$-directionaligns the final dispersion relation with that of the Ising model when $M = L/2$.

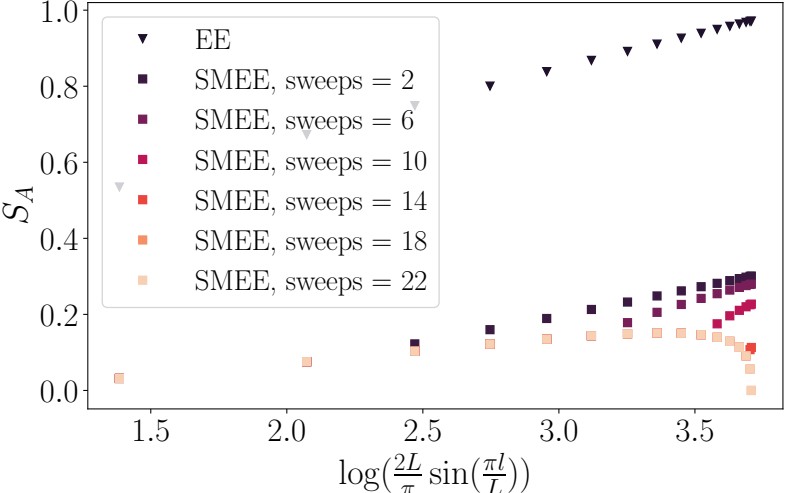

Figure 6: Entanglement entropy (EE) and Stabilizer-minimized Entanglement Entropy (SMEE) at the critical point of Eq. 20. Data are obtained for a fixed length of the system $L = 64$, varying the size of the partitions $A$, $B$ ($\ell_A = l$, $\ell_B = L - Sl$) from $l = 2$ to $l = 32$.

This shows how, at the critical point, the system Hamiltonian can be expressed as a sum of two or three commuting terms, acting on distinct sublattices (up to bundary terms). Each eigenstate can then be mapped onto a direct product of wave functions leaving on the distinct sublattices. This transformation is composed of sequences of SWAP gates, and it thus a Clifford operation.

This implies that, for partition sizes of $L/2$ and $L/3, 2L/3$ respectively, the two models will feature an SMEE resembling exactly that of an Ising model of effective size $L/2$ and $L/3$. Under the assumption that the local minimization procedure we utilize will return that (an assumption that we will scrutinize in detail later), this consideration immediately lends itself to the following predictions: *i)* for $\ell < L/p$, SMEE will scale logarithmically, with an effective central charge that is $1/2$ or $1/3$ of the original one; *ii)* exactly at $L/p$, the SMEE will vanish; *iii)* the same patter will be repeated for all partitions, up to boundary and convergence effects. Moreover, we note that mutual SRE in real space is going to be negative for these cases: this can be seen by noticing that the original transformation can be seen as a correlation acting non-trivially in $A$ and $B$ irrespectively of their distances.

Below, we scrutinize the predictions above against numerical computations.

## 6.1 Magic scaling in cluster Ising models and stabilizer disentangling

Since it is possible to partially disentangle the critical ground state of cluster Ising models with a global Clifford operation, the question we address here is: can this operation be retrieved via local stabilizer disentangling?

It is not automatic that local stabilizer disentangling would be able to find the duality transformation described above. Indeed, the procedure could in principle get stuck into some local minima (since, ultimately, it is made to optimize entanglement locally), and be unable to retrieve the fact that part of the chain can completely decouple from the other. One would then need a Monte Carlo-like procedure to obtain the disentangling operation.

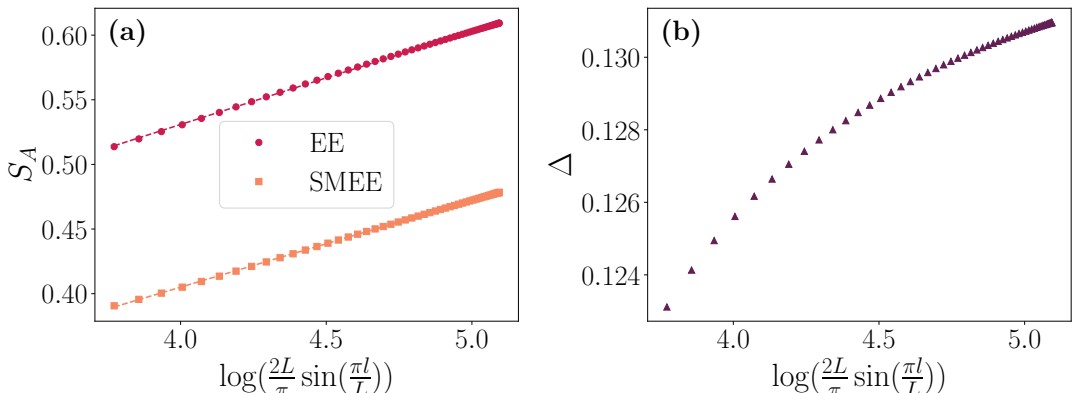

Figure 7: Stabilizer disentangling at the critical point ($D = 0.1$, $h = 0.9$) of Cluster Ising model in Eq. 21. Data are obtained for a fixed length of the system $L = 256$, varying the size of the partitions $A$, $B$ ($\ell_A = l$, $\ell_B = L - l$) from $l = 20$ to $l = 128$. **(a)** Scaling of Entanglement Entropy (EE) and Stabilizer-Minimized Entanglement Entropy (SMEE) with subsystem size $\ell$. Dashed lines represent the fitted curves, with both curves sharing the same slope. **(b)** Scaling of Entropy gain $\Delta$ with subsystem size $\ell$. **(b)** Scaling of entropy gain $\Delta$. It exhibits sublogarithmic scaling; however, due to the very small scale on the y-axis.

In order to have a qualitative picture of this, in Fig. 6, we depict the EE and SMEE for the critical point. One can observe that, differently from all previous cases, there is a strong sweep dependence here: few sweeps systematically overestimate the SMEE.

Most importantly, once converged, the data are perfectly consistent with the observation above: there is a perfect disentangling at $l = L/2$, signaling that simple local Clifford disentangling is able to capture the duality property described above. We note that this is not happening for arbitrary parameters across the transition line: for instance, at $D = 0.1$, full disentangling does not take place, as exemplified in Fig. 7.

To further corroborate the efficiency of disentangling, we consider the Cluster Ising model in Eq. 22, which features instead a tripartite unit cell over duality. The corresponding results are depicted in Fig. 8. As can be seen in both plots (representing different volumes), the SMEE is a strongly non-monotonous function of the partition size. In particular, it becomes compatible with 0 at $L/3$, in perfect agreement with the exact disentanglement expected at convergence. Interesting, we are unable to see the same behavior at $2L/3$ at large volumes (we observe it only at $L = 24$): we believe this may be due to the fact that the transformation has large (3-site) support, so a minimization based on two-body gates only may not be enough for full disentangling.

As a final confirmation of our theory, in Fig. 9, we plot the SMEE obtained from the cluster Ising models, with the EE and SMEE of the of the Ising model defined on a third of its volume. We observe an almost perfect quantitative agreementn (no adjustable parameters) between the Ising SMEE and the Cluster Ising SMEE of a larger system, corroborating the fact that local Clifford circuits are indeed able to fully capture the model duality at the full wave function level. Moreover, in the cluster Ising model at criticality, $L_{AB} < 0$. This completes the picture we proposed: for CAMPS to showcase systematic improvement over MPS, a negative mSRE and the presence of non-local stabilizer information is key.

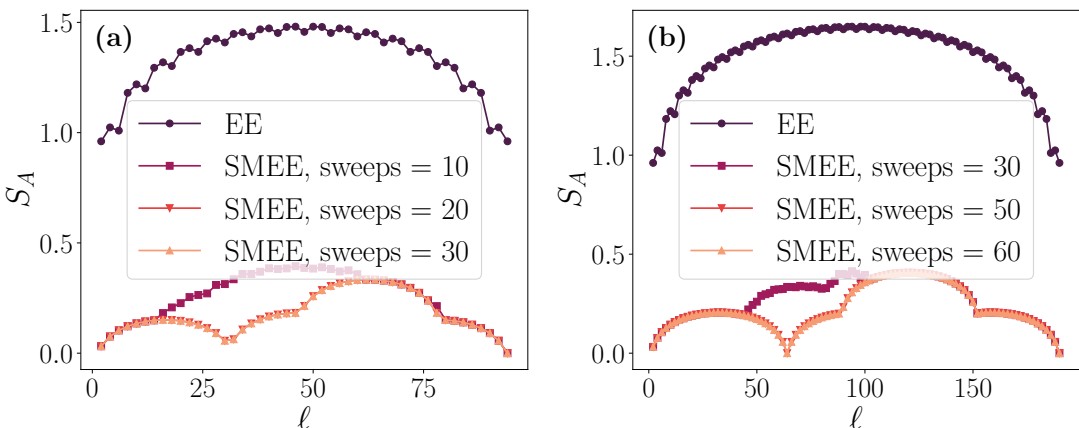

Figure 8: Entanglement entropy (EE) and Stabilizer-minimized Entanglement Entropy (SMEE) at the critical point of Cluster Ising model in Eq. 22. **(a)** System size $L = 96$; **(b)** System size $L = 192$.

## 7 Conclusions

We have scrutinized the possibility of systematically disentangle ground states of quantum spin chains using local Clifford operations. From the complexity viewpoint, the procedure we described can be thought of as a lowering of entanglement complexity, while keeping magic complexity invariant. More specifically, the expectation value of any observable with low Pauli weight can be efficiently computed on an MPS with lower bond dimension, thus reducing the computational cost.

Within this framework, we have shown how the large corner of the Hilbert space comprising matrix product states features very diverse sets. Some states are local-Clifford disentanglable, that is, once system size increases, the bond dimension required to describe a given state is parametrically smaller than that of an MPS. This implies that, to simulate such states, hybrid methods such as Clifford-augmented MPS are systematically better than traditional MPS (they trade a systematic cost for a gain that improves with size). At specific points in parameter space, such states can even become fully disentanglable. Another class of states is instead made of states that cannot be Clifford disentangled apart from a constant contribution.

We have elucidated the origin of such a mechanism utilizing mutual Stabilizer Rényi entropies. Despite not being measures of magic for mixed states, the latter still serve as a quantifier of localization in stabilizer space. Combining qualitative reasoning with exactly solvable examples, we have argued that the key feature for a state to be stabilizer disentangled resides in the sign of the mutual Stabilizer Rényi entropy: if negative, it captures the fact that there are non-local correlations (entanglement) that can be systematically diminished via the application of Clifford operations only.

Our work paves the way to a series of extensions, and leaves open questions. First of all, it calls for a deeper (field) theoretical understanding of mSRE. While those have already been shown to be superior to full state magic in determining critical behavior in a number of cases, they still lack a generic relation to continuum description. Based on the results reported above, understanding the latter would be crucial to better identify the relation between the hybrid complexity classes defined above and physical properties - and, ultimately, defined for which classes of phenomena lowering complexity with CAMPS can work. Similarly, it would be important to understand the role of mSRE in out of equilibrium dynamics. The fact that

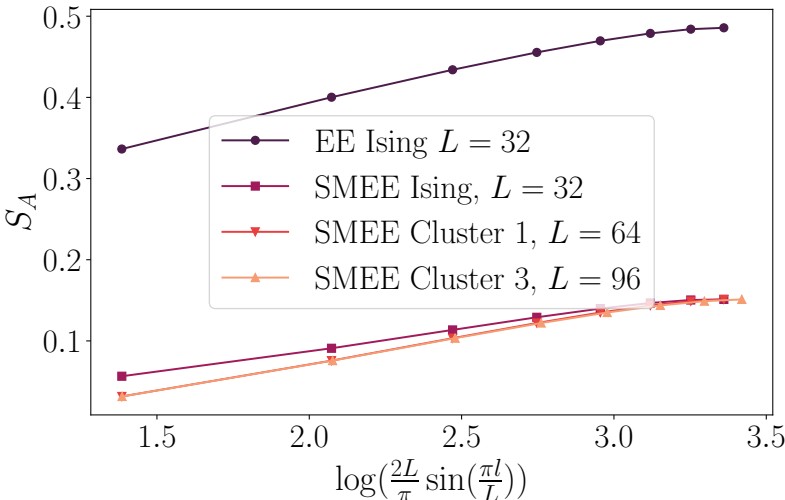

Figure 9: Comparison of Entanglement Entropy in Ising and Cluster Ising models after Stabilizer Disentangling. Data are shown for a fixed system size $L$, indicated in the legend, varying the sizes of the partition $A$, $B$ ($\ell_A = l$, $\ell_B = L - l$) from $l = 2$ to $l = 16$. Both EE and SMEE exhibit the expected logarithmic scaling behavior, and, for different models, SMEEs are compatible.

in those scenarios, improved performances of CAMPS have been so far less promising that for ground states, may be explained by the fact that in such situations, it is very unlikely to observe states with negative mSRE.

At a broader theoretical level, it would be extremely interesting to formulate measures of computational complexity that are hybrid between entanglement and magic - and can thus lower bound the complexity of representing a state using either tensor networks or Pauli tableaus, and establish connections between tensor networks are alternative approaches [68]. Understanding such hybrid complexity would be crucial to address the capability of other classes of Clifford-augmented tensor networks, including tree-tensor and projected-entangled paired states, that are better suited to represent higher-dimensional states. Our proposal to attack this targeting participation entropy in stabilizer subspace could be checked in all such tensor network settings utilizing Pauli-Markov chains, and may stimulate the search for alternative quantities to define hybrid complexity. Finally, it may also be interesting to explore alternatives to systematic disentangling, e.g., including an element of stochasticity, to further decrease entanglement.

## Acknowledgments

We thank B. Béri, R. Fazio, G. Fux, A. Hamma, T. Haug, R. Nehra, M. Qin and H. Timsina for inspiring discussions.

**Funding information**    P.S.T. acknowledges funding by the Deutsche Forschungsgemeinschaft (DFG, German Research Foundation) under Germany's Excellence Strategy – EXC-2111 – 390814868. M.D. was partly supported by the QUANTERA DYNAMITE PCI2022-132919, by the EU-Flagship programme Pasquans2, by the PNRR MUR project PE0000023-NQSTI, by the PRIN programme (project CoQuS), by the MIUR Programme FARE (MEPH), and by the ERC Grant WaveNets.

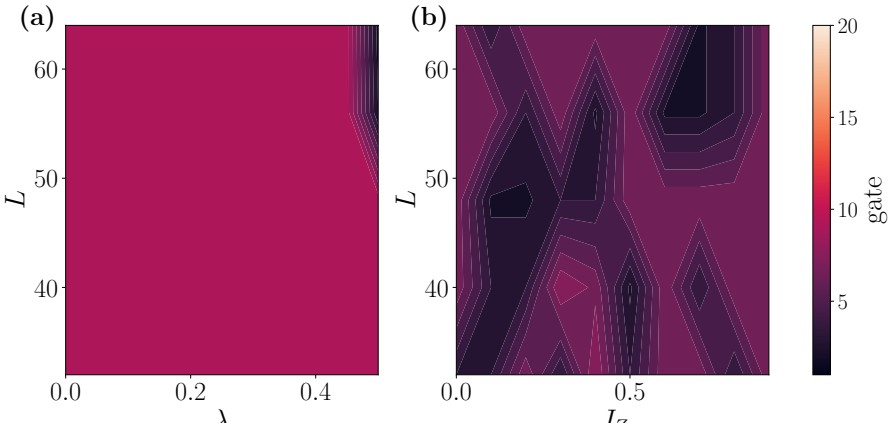

Figure 10: Illustration of gate selection during the first sweep of the Stabilizer Disentangling protocol, across various parameters and system sizes. The gate selected in the process is uniform in the bulk of the chain. **(a)** Results for the TCI model; **(b)** Results for the XXZ model.

**Note added**  After this work appeared on arXiv, a related manuscript by Fan and coworkers was posted (Phys. Rev. B 111, 085121 (2025)). While models and protocol differ, their conclusions on different performances on CAMPS wave functions are in agreement with our findings.

# A  Gate choosing

In this section, we provide detailed descriptions of the gate selection process employed by the Stabilizer Disentangling protocol. Fig. 10 shows the results for two distinct models: the Tricritical Ising model in Fig. 10a and the XXZ model in Fig. 10b. In both cases, onvergence was independent of the number of sweeps, with a single sweep being sufficient to achieve the minimum SMEE. Interestingly, even without constraints on gate selection, the disentangling protocol uniformly selects the same gate in the bulk of the chain, consistent with expectations based on the homogeneous structure of the ground states. For this reasons, the single gate showed in the colormaps in Fig. 10 corresponds to the one selected in the bulk of the chains during the first sweep of the minimization. In Fig. 10a we observe that for the Tricritical Ising model case, where groundstates belong to the LCD of the Hilber space (both for critical point and area-law phases), gate selection remains independent of system size $L$ and parameter $\lambda$. In contrast, for XXZ model, whose groundstates fall within the nLCD sector, the gate selections shows some variations across different values of the parameter $J_z$ in the critical phase. Although the gate choice is not identical for each combination of $J_z$ and $L$, the protocol still explores only a limited subset of $\tilde{C}_2$ gates for minimization.

# B  Estimation of critical point in cluster Ising models

We consider the Hamiltonian in Eq. 21 and estimate the critical value of the parameter $h$ for a fixed $D = 0.1$ by analyzing the system's entanglement entropy. Our results are depicted in Fig. 11. On the left panel, we show the entropy divided by $\log(L)$ for various volumes. This indicates rather sharply that, in the vicinity of 0.9, a critical scaling is observed (another regime that resemble criticality also emerges at larger $h$, but we do not consider it here).

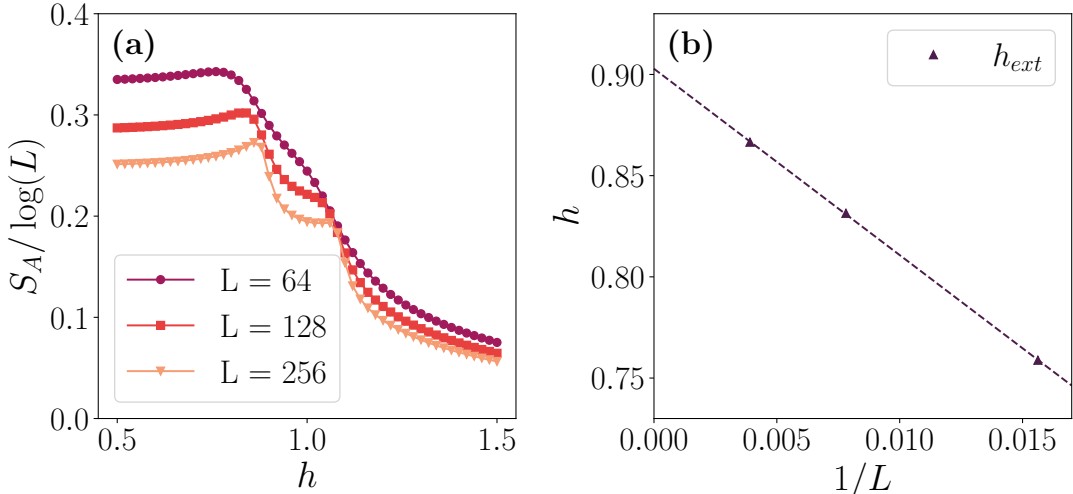

Figure 11: Estimation of the critical point of the Hamiltonian in Eq. 21, for fixed parameter $D = 0.1$. **(a)**: Scaling of Entanglement Entropy (EE) with parameter $h$ for various system sizes. **(b)** Size scaling for the estimation of the critical point $h_c$. The points correspond to the positions of the peak in panel (a), the dashed line represnts the linear fit. The estimated intercept is $h_c = 0.903 \pm 0.001$.

In order to estimate the position of the critical point, we utilize a fine grid in parameter space, and plot the position of the maximum versus inverse volume in the right panel. We observe a trend that is compatible with linear scaling, and extract $h_c = 0.903 \pm 0.001$.

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
