# Peer review of "Stabilizer disentangling of conformal field theories"

_SciPost Physics, doi:SciPost Phys. 18, 165 (2025)_

## Round 1 · Referee Report · Anonymous (Referee 1) · 2025-2-18

Strengths

The effect of sequential disentanglement on the one-dimensional critical states is evaluated by means of the MPS formalism.

Weaknesses

The link between the structure (or definition) of the Hamiltonian and the corresponding ground state property is not straight forward.

Report

The effect of the local and sequential applications of Chlifford gates on the entanglement entropy is examined for the critical parameter case of several one-dimensional quantum models. The numerical investigation is performed for the ground state of finite size systems, which is represented by the matrix product state created by the DMRG method. It turned out that there is a group of states where the suppression of the entanglement entropy increases with the system size. In this manner, it is numerically confirmed that critical ground states can be classified into at least two groups. Theoretical origin of this phenomena is discussed. Definitions of calculated values are given clearly, and the explanations are well structured. As an article which provides new view points on the critical ground state, I recommend the publication of this article even as it is.

Recommendation

Publish (easily meets expectations and criteria for this Journal; among top 50%)

  • validity: good
  • significance: high
  • originality: high
  • clarity: good
  • formatting: good
  • grammar: perfect

Author:  Martina Frau  on 2025-03-27  [id 5321]

(in reply to Report 1 on 2025-02-18)

We thank the Referee for their careful evaluation of the manuscript and for recognizing the potential impact of our work to understanding entanglement entropy suppression in critical ground states and its classification into distinct groups. Their recommendation for publication as it is greatly encourages us. We are pleased that the explanations and definitions in our manuscript were clear and well-structured.

---

## Round 1 · Referee Report · Anonymous (Referee 2) · 2025-2-23

Strengths

  • interesting results on the interplay between magic and disentangling power

Weaknesses

  • unclear what are the main conclusions

Report

The paper studies the possibility of disentangling ground state of one dimensional systems that are conformal invariant through local Clifford disentangler. The question is timely and interesting; however, I am not sure to follow exactly what the authors want to claim. I understand that the disentangling power is related to the amount of magic, but then I don't really see a clearly presented relation between short range magic and long-range one. The authors introduce m_chi=2 which it is really difficult to understand what its meaning. Why chi = 2 and not chi = 1 or 6? Disentangling should be about decomposing log scaling entanglement to area law. Moreover, given that these are CFTs, one should also study how critical proprieties change by increasing chi and disentangling power. I find the paper quite drafty, with no clear sense of what are the main statements and physical implications.

Requested changes

  • clarify the main messages regarding magic and entangling power, clarify the role of m_chi= 2
  • study critical proprieties (for example the spectral gap) as function of bond dimension and disentangling power

Recommendation

Ask for major revision

  • validity: high
  • significance: good
  • originality: good
  • clarity: good
  • formatting: excellent
  • grammar: excellent

Author:  Martina Frau  on 2025-03-27  [id 5320]

(in reply to Report 2 on 2025-02-23)
Category:
remark
answer to question
reply to objection

Dear Referee,
we thank you for your careful evaluation of the manuscript. We've attached a detailed point-by-point reply to the report including additional graphs.
Yours sincerely,
the authors

Attachment:

Reply_Referee_2.pdf

---

## Round 2 · Referee Report · Anonymous (Referee 2) · 2025-4-7

Report

i thank the authors for their replies and iteration of the manuscript. the paper is ready for publicatio now.

Recommendation

Publish (easily meets expectations and criteria for this Journal; among top 50%)

---

## Round 2 · Referee Report · Anonymous (Referee 1) · 2025-4-15

Strengths

(copy from the previous report.) The effect of sequential disentanglement on the one-dimensional critical states is evaluated by means of the MPS formalism.

Weaknesses

(copy from the previous report) The link between the structure (or definition) of the Hamiltonian and the corresponding ground state property is not straight forward.

Report

As I wrote in the previous report, the submitted previous manuscript (ver.1) was already publishable. The current resubmitted manuscript (ver.2) contains sufficient clarifications to the comments of the other referee. Thus there is no reason of blocking publication of this article.

Recommendation

Publish (easily meets expectations and criteria for this Journal; among top 50%)

---

## Round 2 · List of Changes

• We double-checked Arxiv references for updates, in particular, we updated Refs.[16,17,48,50,52]
and the Note added paragraph
• We added a DOI in Refs. [1-4,7-10,12,13,18-23,25,36,40,41,49,58]
• We changed a paragraph in the introduction to clarify the summary of our main results,
including a description of the connection between mχ=2 and the disentangling power.
• We added a paragraph in Section 2.5 to motivate the choice of employing χ = 2 to
measure the magic that cannot be removed by local operation.

---

## Editorial Decision

published